# Atomically dispersed Iridium on $Mo_2C$ as an efficient and stable alkaline hydrogen oxidation reaction catalyst

Jinjie Fang[1,4], Haiyong Wang[1,4], Qian Dang[1,4], Hao Wang[1], Xingdong Wang[1], Jiajing Pei[1], Zhiyuan Xu[1], Chengjin Chen[1], Wei Zhu [1], Hui Li[1] ✉, Yushan Yan [2] ✉ & Zhongbin Zhuang [1,3] ✉

Hydroxide exchange membrane fuel cells (HEMFCs) have the advantages of using cost-effective materials, but hindered by the sluggish anodic hydrogen oxidation reaction (HOR) kinetics. Here, we report an atomically dispersed Ir on $Mo_2C$ nanoparticles supported on carbon ($Ir_{SA}$-$Mo_2C$/C) as highly active and stable HOR catalysts. The specific exchange current density of $Ir_{SA}$-$Mo_2C$/C is $4.1\,mA\,cm^{-2}_{ECSA}$, which is 10 times that of Ir/C. Negligible decay is observed after 30,000-cycle accelerated stability test. Theoretical calculations suggest the high HOR activity is attributed to the unique $Mo_2C$ substrate, which makes the Ir sites with optimized H binding and also provides enhanced OH binding sites. By using a low loading ($0.05\,mg_{Ir}\,cm^{-2}$) of $Ir_{SA}$-$Mo_2C$/C as anode, the fabricated HEMFC can deliver a high peak power density of $1.64\,W\,cm^{-2}$. This work illustrates that atomically dispersed precious metal on carbides may be a promising strategy for high performance HEMFCs.

Utilizing hydrogen energy presents a viable solution to reduce the dependence on fossil fuels and tackle environmental concerns[1,2]. In the pursuit of establishing hydrogen economy, fuel cells play crucial roles[3,4]. Hydroxide exchange membrane fuel cells (HEMFCs), which are analogs of proton exchange membrane fuel cells (PEMFCs) but using hydroxide exchange membranes, are promising due to their cost-effectiveness coming from the possibility to use platinum group metals (PGMs) free cathode electrocatalysts (e.g., Ag, Mn-Co oxides and Fe-N-C) and Ni-based bipolar plates[5,6]. However, high-performance anodes for HEMFCs still rely on PGMs[7]. Moreover, the pH effect on PGMs (e.g., Pt, Rh, Pd and Ir) leads to a significant slowdown of the hydrogen oxidation reaction (HOR) kinetics by approximately 2-3 orders of magnitude when changing the electrolyte from acid to alkaline, which results in high PGMs loadings on anode of HEMFCs[8]. Consequently, the anode cost in HEMFC substantially increases. Additionally, the poor stability of commonly used Pt-based catalysts necessitates higher catalyst loadings to enable a stable HEMFC, further

increasing costs[9]. Hence, the development of highly efficient and stable catalysts with lower loading of PGMs for alkaline HOR is a pivotal target for the ongoing advancement of HEMFCs.

Significant efforts have been made to develop highly efficient catalysts for alkaline HOR[10] and have shown that by tuning the microenvironment of active sites, the binding energies of HOR intermediates could be adjusted to improve the HOR kinetics[11]. Alloys, heterostructures and decorated catalysts have been reported with elevated HOR performance compared with the pristine metal catalysts by adjusting the H and OH binding energies[12–15]. Ir, which demonstrates enhanced OH bindings than Pt, has been shown to be a promising candidate for HOR in alkaline. Ir alloy (e.g., IrNi, IrMo and IrRu)[16–18] and heterostructures (e.g., $Ir/MoS_2$ and $Ir/WO_x$)[19,20] have been reported with enhanced HOR activities, even comparable to the state-of-the-art PtRu/C catalysts. The catalyst HOR stability is also strongly affected by the microenvironment of active sites[9,21]. Recent studies ascribed the poor stability of commercial carbon-supported Pt nanoparticle (NP)

[1]State Key Lab of Organic–Inorganic Composites and Beijing Advanced Innovation Center for Soft Matter Science and Engineering, Beijing University of Chemical Technology, Beijing, China. [2]Department of Chemical and Biomolecular Engineering, University of Delaware, Newark, DE, USA. [3]Beijing Key Laboratory of Energy Environmental Catalysis, Beijing University of Chemical Technology, Beijing, China. [4]These authors contributed equally: Jinjie Fang, Haiyong Wang, Qian Dang. ✉e-mail: hli@mail.buct.edu.cn; yanys@udel.edu; zhuangzb@mail.buct.edu.cn

catalysts to the local carbon support corrosion[22,23]. Especially, PGMs can catalyze the carbon corrosion, making it especially serious at the interface of Pt NP and carbon support. Thus, a more stable substrate is desired to load the PGMs with less corrosion. Alternative supports, such as oxides, carbides, and nitrides, have been reported to improve the stability[21,24]. For example, $TiO_2$-$RuO_2$ was used as the support for Pt nanoparticles, and enhanced fuel cell stability was achieved[25]. Another method is to use a buffer to separate PGMs and the carbon substrates. For example, Dekel et. al.[26] reported $Pd/CeO_2$-C with enhanced stability, by using $CeO_2$ as buffer substrate to load Pd nanoparticles, which could prevent the direct contact between Pd sites and the carbon support and thus mitigate the local carbon corrosion[27]. This work indicates that a rationally designed catalyst with a tailored microenvironment could also achieve high activity and stability.

Atomically dispersed metal catalysts show great potential for high-performance electrocatalysts due to their high atom-utilization and special microenvironment around the metal sites[28,29]. However, the commonly used M-$N_4$-C type atomically dispersed catalysts do not show satisfactory performance for HOR, especially in HEMFC devices, because their significantly different metal microenvironments compared with those in pristine metal NPs may lead to improper binding to the HOR intermediates, such as adsorbed H (*H) and adsorbed OH (*OH)[30-32]. As a result, if a properly selected substrate can both host the atomically dispersed PGMs and achieve the optimized metal site microenvironment, it may simultaneously bring high HOR activity and stability.

Herein, we report an atomically dispersed Ir on $Mo_2C$ NPs supported on carbon (denoted as $Ir_{SA}$-$Mo_2C$/C) as a highly active and stable HOR catalyst in alkaline electrolyte. $Mo_2C$ NPs are used as the hosts for Ir atoms. The $Mo_2C$ shows Pt-like electronic structure, enabling the guest Ir to have a unique binding property with intermediates[33-35]. $Mo_2C$ also has high stability and strong interaction with PGMs, making Ir atoms atomically dispersed in the hexagonal $Mo_2C$ matrix. The obtained $Ir_{SA}$-$Mo_2C$/C catalyst achieves excellent activity for HOR, indicated by the high specific exchange current density of 4.1 mA $cm^{-2}_{ECSA}$ and mass activity at 50 mV versus reversible hydrogen electrode (RHE, the same hereafter) of 17.9 A $mg^{-1}_{Ir}$, which are 10 and 12 times that of Ir/C, and 2.6 and 4.3 times that of the current state-of-the-art PtRu/C, respectively. By using a low anode PGM loading of only 50 $\mu g_{Ir}$ $cm^{-2}$, the fabricated $Ir_{SA}$-$Mo_2C$/C HEMFC can deliver a high peak power density (PPD) of 1.64 W $cm^{-2}$. The $Ir_{SA}$-$Mo_2C$/C illustrates high stability as well. It can work stably in a 120-h continuous operation and 30,000 cycles of accelerated durability test (ADT). Density functional theory (DFT) calculations reveal that the d-orbital of $Ir_{SA}$-$Mo_2C$ is much different from that of the HOR inactive M-$N_4$-C type atomically dispersed catalysts, and the high activity of $Ir_{SA}$-$Mo_2C$/C comes from the enhanced hydroxide binding as well as the optimized hydrogen binding. The present study demonstrates the advantages of the atomically dispersed metal on carbides in electrocatalysis.

## Results and Discussion
### Catalyst synthesis and characterization
The $Ir_{SA}$-$Mo_2C$/C catalyst was synthesized by a two-step approach (Fig. 1a). Firstly, $MoO_x$ was grown on pre-oxidized carbon black via the hydrolysis of $MoCl_5$. Supplementary Fig. 1 shows the transmission electron microscopy (TEM) image and the corresponding energy-dispersive X-ray spectroscopy (EDS) mappings. It demonstrated that Mo was homogeneously dispersed on the carbon support. The broad shoulder peak in powder X-ray diffraction (PXRD) pattern (Supplementary Fig. 2) suggested the formation of amorphous $MoO_x$. Subsequently, Ir was introduced by impregnation, followed by calcination at 900 °C under nitrogen environment. Under high temperature, carbon atoms would diffuse from carbon black to $MoO_x$ and produce the $Mo_2C$ NPs[36]. At the same time, Ir was atomically dispersed on it to form

the $Ir_{SA}$-$Mo_2C$/C catalyst because of the strong interaction between Ir atoms and $Mo_2C$ host.

Figure 1b shows the PXRD pattern of the $Ir_{SA}$-$Mo_2C$/C. All the diffraction peaks were assigned to hexagonal $Mo_2C$ without any signal from Ir-derived species, such as Ir and $IrO_2$. The TEM image of $Ir_{SA}$-$Mo_2C$/C (Fig. 1c) clearly shows the well-dispersed NPs with an average size of 2.4 nm (inset of Fig. 1c) on the carbon supports. The aberration-corrected high-angle annular dark-field scanning TEM images (AC-HAADF-STEM, Fig. 1d) illustrated two sets of lattice with spacing of both 0.23 nm and angle of ca. 57°, which could be assigned to the (101) and ($\bar{1}$01) facets of hexagonal $Mo_2C$[37]. Figure 1e exhibits the line intensity profile of the two areas in Fig. 1d. Isolated dots with particularly high intensity were observed, which was assigned to Ir atoms because of the higher atomic number of Ir than Mo[15]. It was confirmed that Ir substituted the Mo sites in $Mo_2C$, and Ir were atomically isolated. Furthermore, the EDS mapping (Fig. 1f) verified the uniform dispersion of Ir and Mo. The Ir/Mo molar ratio in $Ir_{SA}$-$Mo_2C$/C was ca. 1:11 from EDS spectra (Supplementary Fig. 3), which was consistent with the ratio of ca. 1:10 tested by inductively coupled plasma optical emission spectra (ICP-OES). The Ir loading in $Ir_{SA}$-$Mo_2C$/C was ca. 3.1 wt % from ICP-OES.

Then, the electronic structures and coordination environments of Ir in $Ir_{SA}$-$Mo_2C$/C were further investigated by X-ray absorption spectroscopy (XAS). The Ir $L_3$-edge X-ray absorption near-edge structure (XANES, Fig. 2a) of $Ir_{SA}$-$Mo_2C$/C exhibited the white line intensity between that of Ir foil and $IrO_2$, revealing that the valence state of Ir in $Ir_{SA}$-$Mo_2C$/C was between that of Ir and $IrO_2$[38]. The same conclusion was obtained from the Ir 4f X-ray photoelectron spectroscopy (XPS, Fig. 2d). The $4f_{7/2}$ peak of Ir/C could be deconvoluted into three peaks: $Ir^0$ at 61.5 eV, $Ir^{4+}$ at 62.4 eV and its satellite peak at 64.0 eV[39,40]. The existence of the $Ir^{4+}$ species is likely from the surface oxidation of nanoparticles[41]. For $IrO_2$, only the $Ir^{4+}$ species and satellite peak were observed. The peak of $Ir_{SA}$-$Mo_2C$/C was located in between of those for $Ir^0$ and $Ir^{4+}$, indicting its medium oxidation state. Supplementary Fig. 4 shows the Mo 3d XPS spectra of $Ir_{SA}$-$Mo_2C$/C and $Mo_2C$/C, indicating the proportion of $Mo^{6+}$ species increased after introducing Ir. The Fourier-transformed $k^2$-weighted extended X-ray absorption fine structure (EXAFS, Fig. 2b) shows that $Ir_{SA}$-$Mo_2C$/C exhibits a major peak at ca. 1.75 Å, which comes from the Ir−C path. No Ir−Ir path (2.52 Å, reference Ir foil) or Ir−O path (1.62 Å, reference $IrO_2$) was observed. We further performed the wavelet transform EXAFS (Fig. 2e) analysis to distinguish back-scattering atoms in the k-space. The $Ir_{SA}$-$Mo_2C$/C revealed only one intensity maximum located at 4.1 $Å^{-1}$, which was different from that of $IrO_2$ (5.8 $Å^{-1}$) and Ir foil (11.4 $Å^{-1}$), suggesting the unique Ir−C scattering path in $Ir_{SA}$-$Mo_2C$/C[31]. Furthermore, the FT-EXAFS fitting in the R-space (Fig. 2c) and the k-space (Supplementary Fig. 5) suggested the Ir−C scattering path of 2.10 Å after phase correction with the coordination number of ca. 3.1, which was reasonably close to the Mo−C path in $Mo_2C$ located at 2.11 Å with corresponding coordination number of 3. The fitting curves for Ir foil and $IrO_2$ were shown in Supplementary Fig. 6, and results were summarized in Supplementary Table 1. Thus, the structure of $Ir_{SA}$-$Mo_2C$ was confirmed as the Ir atoms partially replaced the Mo atoms in the hexagonal $Mo_2C$ lattice and thus coordinated with C.

### High HOR activity of the $Ir_{SA}$-$Mo_2C$/C
The HOR electrocatalytic activity of the $Ir_{SA}$-$Mo_2C$/C was firstly investigated by the rotating disk electrode (RDE) method in $H_2$-saturated 0.1 M KOH using the standard three-electrode system. The commercial PtRu/C, Pt/C and $IrO_2$ were tested at the same condition for comparison. The commonly used carbon-based Ir single atom catalysts (i.e., $Ir_{SA}$-$N_4$-C, synthesized through the procedure reported in refs. 32,42. Ir/C (synthetic procedure shown in Methods, TEM images and XRD pattern shown in Supplementary Fig. 7), and Ir nanocluster on $Mo_2C$ supported on carbon ($Ir_{NC}$-$Mo_2C$/C, synthetic procedure shown in

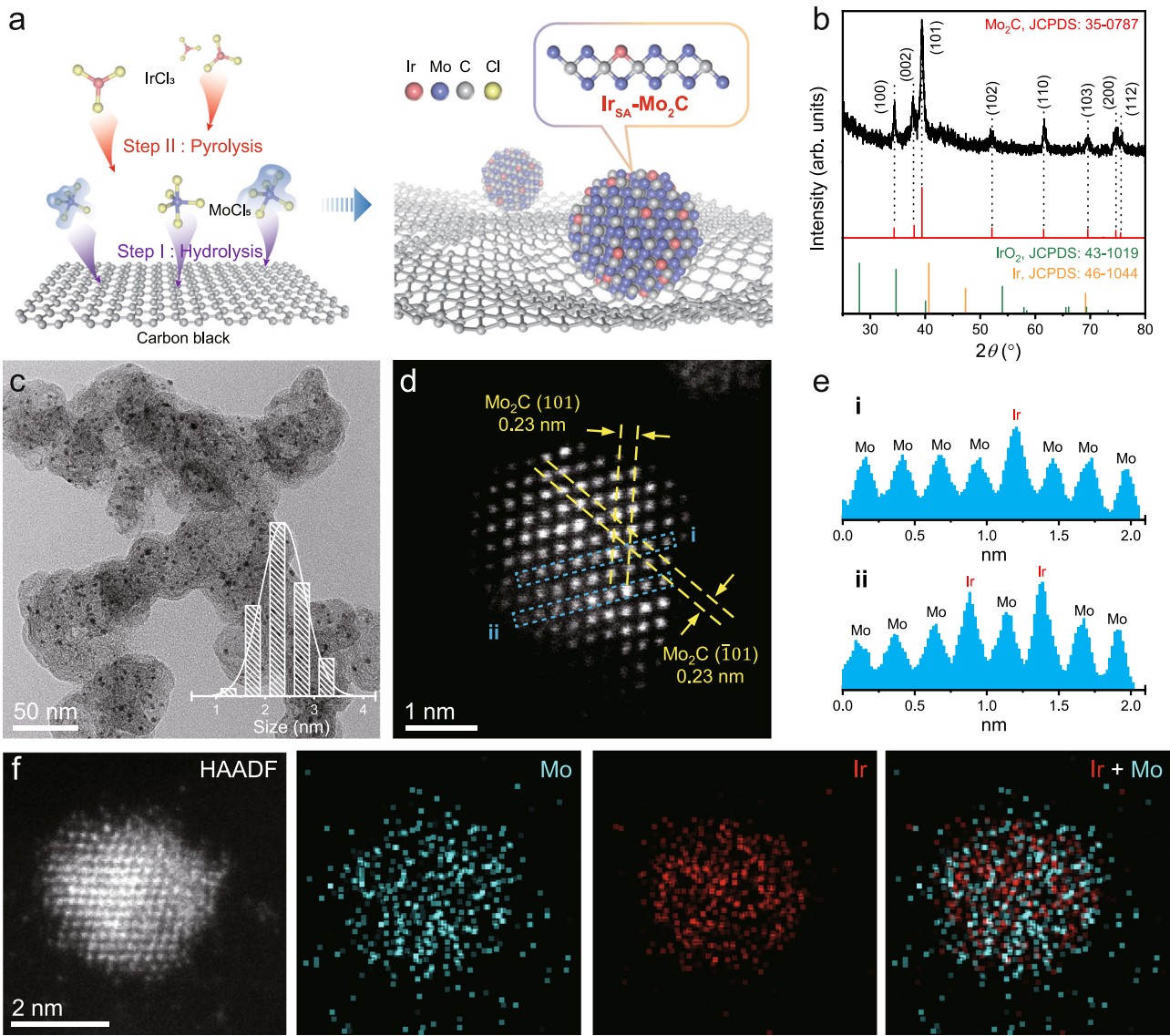

**Fig. 1 | Synthesis of Ir$_{SA}$-Mo$_2$C/C and its characterizations. a** Schematic illustration of the synthesis procedure. **b** PXRD pattern. **c** TEM image. The inset shows the particle size distribution. **d** AC-HAADF-STEM image. **e** Intensity profiles of the line i and ii in (**d**). **f** HAADF-EDS mapping results.

Methods, TEM images and XRD pattern shown in Supplementary Fig. 8) were tested as control samples. Different from the negligible HOR activity of the Ir$_{SA}$-N$_4$-C (Supplementary Fig. 9, similar performance was reported by Q. Wang et al.)[43] the Ir$_{SA}$-Mo$_2$C/C illustrated high HOR performance, indicated by the steeply increased anodic current density (Fig. 3a). The Mo$_2$C/C support showed scarce HOR activity, suggesting that the high performance of Ir$_{SA}$-Mo$_2$C/C came from the Ir sites. And the Ir$_{SA}$-Mo$_2$C/C also displayed higher HOR activity than Ir$_{NC}$-Mo$_2$C/C, indicating the benefit of the monoatomic dispersed Ir in Mo$_2$C. The Ir$_{SA}$-Mo$_2$C/C showed better performance than the other Ir-based catalysts, such as Ir/C and IrO$_2$ (Fig. 3a and Supplementary Fig. 10), as well as the commonly used Pt-based catalysts, such as Pt/C and the state-of-the-art PtRu/C (Fig. 3a).

We further tested the polarization curves of Ir$_{SA}$-Mo$_2$C/C at different rotating speeds (Fig. 3b). By fitting with the Koutecky-Levich equation (Fig. 3c), a slope of 4.70 cm$^2$ mA$^{-1}$ s$^{-1/2}$ was obtained, which was reasonably close to the theoretical value of the 2e HOR process (4.87 cm$^2$ mA$^{-1}$ s$^{-1/2}$) and suggested the anode current mainly derived from HOR[44,45].

In order to quantitatively compare the HOR activity, we calculate the specific exchange current density ($j_{0,ECSA}$) based on the electrochemical surface area (ECSA) and the mass activity at 0.05 V of the catalysts. The ECSA of the catalysts are determined by CO stripping voltammetry, and the same specific charge of 420 μC cm$^{-2}$ is used for all the catalysts (Supplementary Fig. 11 and Supplementary Table 2). The Ir$_{SA}$-Mo$_2$C/C exhibits an ECSA of 156 m$^2$ g$^{-1}_{Ir}$, which was larger than the ECSA values of Ir/C (128 m$^2$ g$^{-1}_{Ir}$), Pt/C (97 m$^2$ g$^{-1}_{Pt}$) and PtRu/C (107 m$^2$ g$^{-1}_{PtRu}$)[46]. The high ECSA was benefited from the atomically dispersion of Ir in Ir$_{SA}$-Mo$_2$C/C, which brought high utilization of Ir. Figure 3d shows the ECSA normalized Tafel plots. By fitting with the Butler-Volmer equation according to Tafel-Volmer pathway[45], the Ir$_{SA}$-Mo$_2$C/C provides the greatest $j_{0,ECSA}$ of 4.1 ± 0.16 mA cm$^{-2}_{ECSA}$ (Fig. 3e), which was 2.6, 7.7 and 10 times as high as that for PtRu/C (1.6 ± 0.07 mA cm$^{-2}_{ECSA}$), Pt/C (0.53 ± 0.041 mA cm$^{-2}_{ECSA}$) and Ir/C (0.41 ± 0.054 mA cm$^{-2}_{ECSA}$), respectively. The measured specific $j_{0,ECSA}$ of PtRu/C, Pt/C and Ir/C reproduced the literature data[13,45,47], implying a reliable evaluation of HOR activity. The Ir$_{SA}$-Mo$_2$C/C also exhibited a high mass activity at 50 mV of 17.9 ± 1.2 A mg$^{-1}_{Ir}$, which was 4.3, 13 and 12 times higher than that of PtRu/C (4.12 ± 0.16 A mg$^{-1}_{PtRu}$), Pt/C (1.37 ± 0.06 A mg$^{-1}_{Pt}$) and Ir/C (1.47 ± 0.08 A mg$^{-1}_{Ir}$), respectively. The more significant improvement of the mass activity is attributed to the higher Ir utilization of Ir$_{SA}$-Mo$_2$C/C. Ir$_{SA}$-Mo$_2$C/C also showed great

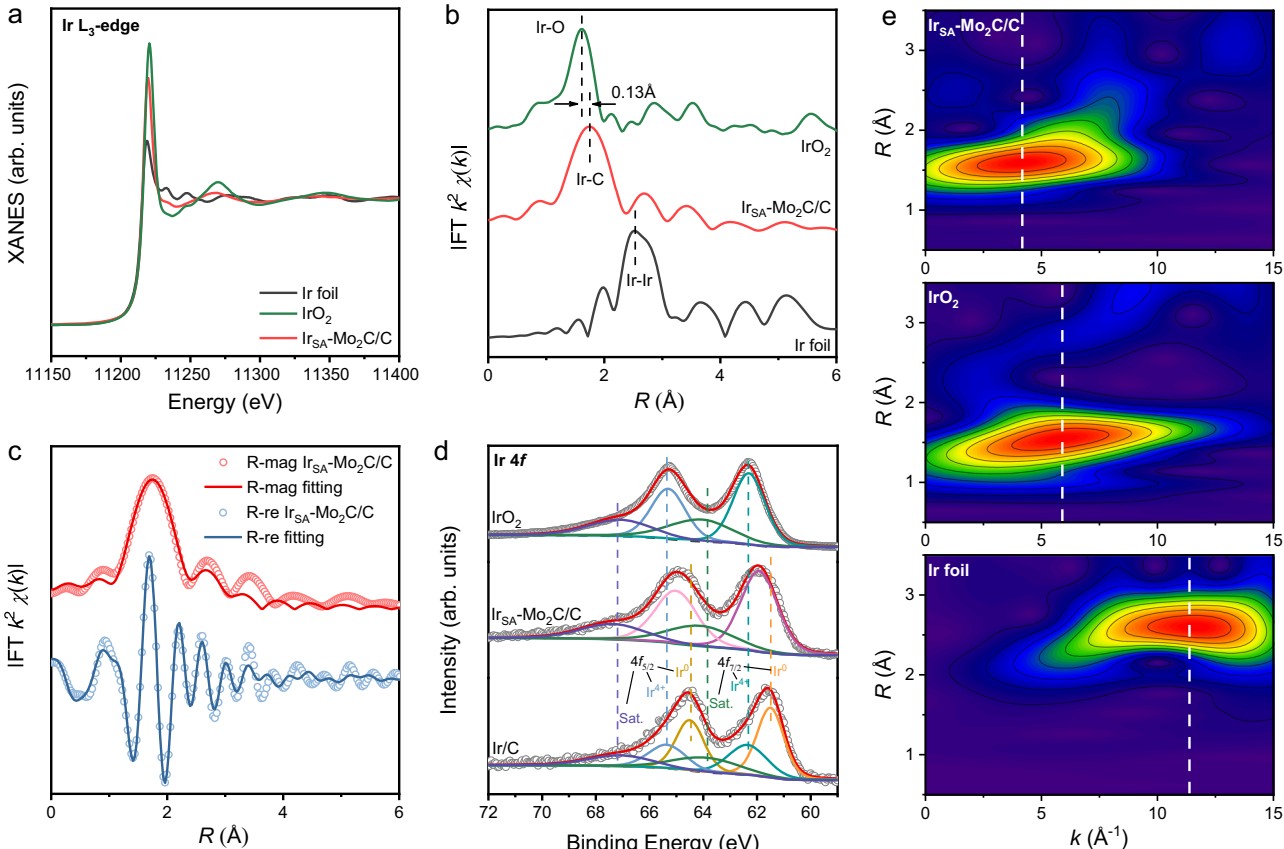

**Fig. 2 | Structural analyzes of Ir$_{SA}$-Mo$_2$C/C. a** Ir L$_3$-edge XANES spectra. The spectra of IrO$_2$ and Ir foil are shown as reference. **b** Corresponding FT $k^2$-weighted EXAFS spectra. **c** FT-EXAFS fitting curves at $R$-space of Ir L$_3$-edge. **d** Ir $4f$ high resolution XPS spectra of Ir$_{SA}$-Mo$_2$C/C, Ir/C and IrO$_2$. **e** Corresponding WT-EXAFS spectra of Ir$_{SA}$-Mo$_2$C/C, IrO$_2$ and Ir foil.

advantages in both specific activity and mass activity compared with the previously reported high-performance HOR catalysts (summarized in Supplementary Fig. 12 and Supplementary Table 3).

Encouraged by the superior HOR activity of the obtained Ir$_{SA}$-Mo$_2$C/C catalyst, we employed it as the anode catalyst to assemble the HEMFCs. The membrane electrode assembly (MEA) was made by using Ir$_{SA}$-Mo$_2$C/C as anode catalyst with a low PGM loading of 0.05 mg$_{Ir}$ cm$^{-2}$ (confirmed by ICP-OES), commercial Pt/C (0.4 mg$_{Pt}$ cm$^{-2}$) as cathode catalyst and the PiperION™ A15 (Versogen) as the hydroxide exchange membrane. The control MEAs were fabricated by only replacing the anode catalyst to PtRu/C, Pt/C and Ir/C with the same loading of 0.05 mg$_{PGM}$ cm$^{-2}$. Figure 3f displays the polarization and power density curves of the as-prepared HEMFCs under H$_2$/O$_2$ condition. The Ir$_{SA}$-Mo$_2$C/C HEMFC exhibited excellent performance, which could deliver high PPD of 1.64 W cm$^{-2}$, and is much better than the HEMFCs using PtRu/C (0.91 W cm$^{-2}$), Pt/C (0.51 W cm$^{-2}$) and Ir/C (0.60 W cm$^{-2}$). The Ir$_{SA}$-Mo$_2$C/C HEMFC achieved a high current density of 1.60 A cm$^{-2}$ at the cell voltage of 0.65 V, which was also higher than that of PtRu/C HEMFC (0.76 A cm$^{-2}$), Pt/C HEMFC (0.43 A cm$^{-2}$) and Ir/C HEMFC (0.49 A cm$^{-2}$). The Ir$_{SA}$-Mo$_2$C/C HEMFC also showed high performance under H$_2$/air(CO$_2$-free) condition, which delivered a PPD of 0.90 W cm$^{-2}$ and reached a current density of 1.11 A cm$^{-2}$ at 0.65 V (Supplementary Fig. 13). Furthermore, we normalized the PPD of these MEAs to the anode catalyst loading to calculate the anode mass activity. As shown in Supplementary Fig. 14 and Supplementary Table 4, Ir$_{SA}$-Mo$_2$C/C HEMFC delivered a high anode mass activity of 32.8 W mg$^{-1}$$_{Ir}$, which was superior to the previously reported HEMFCs.

## Enhanced HOR stability of Ir$_{SA}$-Mo$_2$C/C

The HOR stability of Ir$_{SA}$-Mo$_2$C/C was evaluated by RDE method. Two techniques, chronoamperometry and cyclic voltammetry for ADT, were carried out. Figure 4a exhibits the chronoamperometry response of Ir$_{SA}$-Mo$_2$C/C and the control catalysts in H$_2$-saturated 0.1 M KOH at a constant potential of 50 mV. The current density of Ir$_{SA}$-Mo$_2$C/C maintained ca. 95% of the initial value after continuously operating for 120 h. The polarization curves before and after 120 h operation (Supplementary Fig. 15) showed negligible decay. While for the Ir$_{NC}$-Mo$_2$C/C, Ir/C, Pt/C and PtRu/C, the current density remained only 62 %, 45 %, 35 % and 33 % after a short test of 20 h, respectively. The decay rate of Ir$_{SA}$-Mo$_2$C/C was 0.042 % h$^{-1}$, much smaller than those of Ir$_{NC}$-Mo$_2$C/C (1.9 % h$^{-1}$), Ir/C (2.5 % h$^{-1}$), Pt/C (2.9 % h$^{-1}$) and PtRu/C (3.0 % h$^{-1}$), showing nearly two order of magnitude slowdown of the catalyst decay. Supplementary Fig. 16 and Supplementary Table 5 summarize the decay rate of Ir$_{SA}$-Mo$_2$C/C and the reported HOR catalysts, where Ir$_{SA}$-Mo$_2$C/C showed great advantage.

The stable HOR performance of Ir$_{SA}$-Mo$_2$C/C was also determined by ADT. After scanning at a potential window of 0 ~ 0.5 V with the scanning rate of 0.1 V s$^{-1}$ for 30,000 cycles, Ir$_{SA}$-Mo$_2$C/C did not show any decline (Fig. 4b). By comparison, PtRu/C, Pt/C, Ir/C and Ir$_{NC}$-Mo$_2$C/C exhibited dramatic decline in kinetic current (Fig. 4b and Supplementary Fig. 17). To quantitatively compare the activity loss in ADT, the corresponding exchange currents of these catalysts were summarized in Fig. 4c. Ir$_{SA}$-Mo$_2$C/C maintained its activity in the whole process of ADT without any decline, while PtRu/C, Pt/C, Ir/C and Ir$_{SA}$-Mo$_2$C/C suffered from a decay of 65 %, 76 %,14 % and 45 % in $j_0$ after scanning for

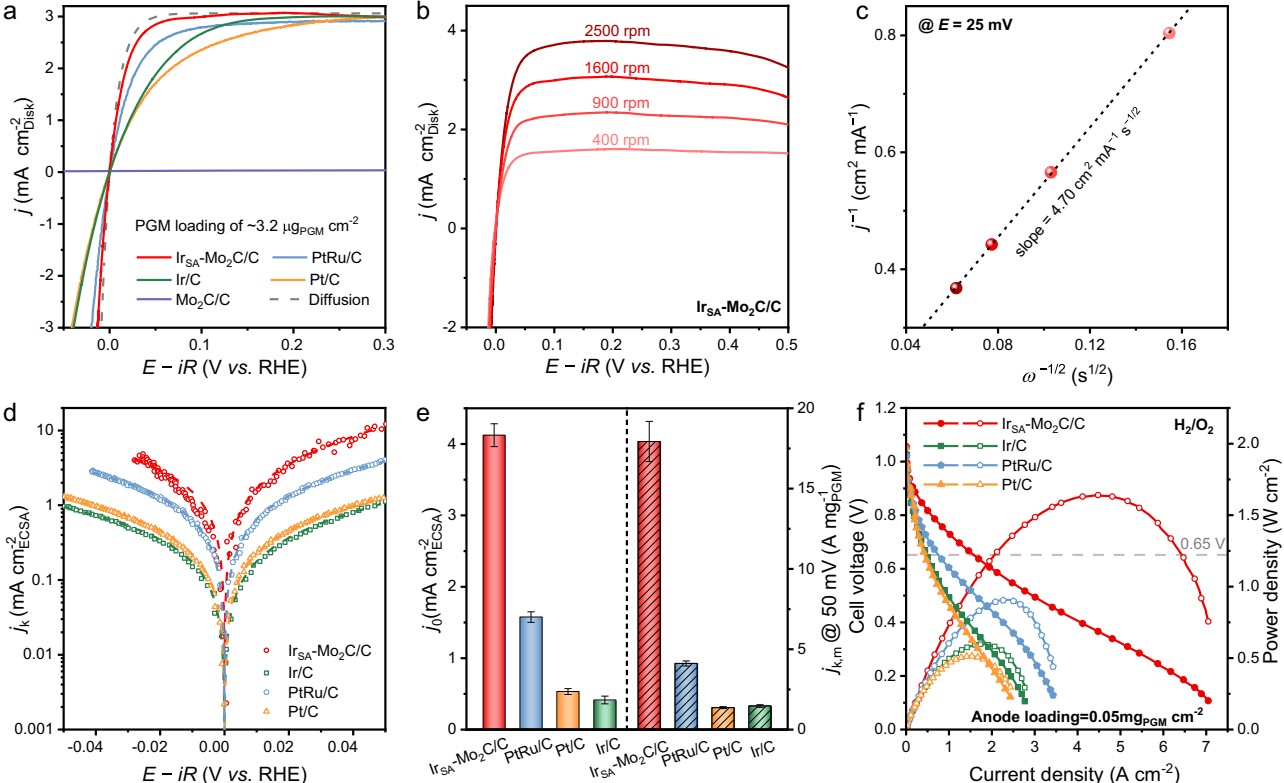

**Fig. 3 | HOR performance. a** Polarization curves of the catalysts in $H_2$-saturated 0.1 M KOH solution with the rotation rate of 1600 rpm and the scan rate of 5 mV s$^{-1}$. The catalyst loading was 3.2 $\mu g_{PGM}$ cm$^{-2}$ for $Ir_{SA}$-$Mo_2C/C$, Ir/C, Pt/C and PtRu/C. The potentials were $iR$ corrected and the $R$ values for $Ir_{SA}$-$Mo_2C/C$, Ir/C, Pt/C and PtRu/C measurements were 36.1 ± 0.10, 35.9 ± 0.12, 36.2 ± 0.10 and 36.0 ± 0.07 Ω, respectively. **b** Polarization curves of $Ir_{SA}$-$Mo_2C/C$ at different rotation rates. **c** The corresponding Koutecky-Levich plot at 25 mV. **d** Tafel plots of the catalysts where the current densities were normalized to their ECSA. The dash line shows the Butler-Volmer fitting curves. **e** The specific exchange current density ($j_{0,ECSA}$) and the mass normalized kinetic current density ($j_{k,m}$) at 50 mV. Error bars are s.d. of at least three sets of experimental repeats. **f** $H_2/O_2$ HEMFC performances using different anode catalysts. The anode PGM loadings were controlled as 0.05 mg$_{PGM}$ cm$^{-2}$. The cathode for all the HEMFCs was 0.4 mg$_{Pt}$ cm$^{-2}$ Pt/C. The cell, anode and cathode humidifier temperatures were 95, 92 and 95 °C, respectively. The anode and cathode were flowed with 1.0 L min$^{-1}$ of $H_2$ and 0.5 L min$^{-1}$ of $O_2$, respectively. The backpressures were 250 kPag for both sides.

10,000 cycles, respectively. The TEM image of $Ir_{SA}$-$Mo_2C/C$ after ADT (Fig. 4d) showed the same structure with almost the same size distribution of NPs (ca. 2.5 nm) compared with the fresh catalyst. The HRTEM image (Supplementary Fig. 18) exhibited lattice spacing of 0.23 nm belonging to the $Mo_2C$ (101). The EDS mapping shows the homogenous dispersion of Ir and Mo atoms in the NPs after ADT. Overall, these results indicated the good stability and the $Ir_{SA}$-$Mo_2C/C$ maintained its microstructure during HOR process.

The $Ir_{SA}$-$Mo_2C/C$ was also stable under the HEMFC working conditions. Supplementary Fig. 19 shows the galvanostatic discharge curve at 500 mA cm$^{-2}$ at 80 °C of the $Ir_{SA}$-$Mo_2C/C$ HEMFC under $H_2/O_2$ condition. It showed negligible voltage loss after 50 h of test.

## Mechanism investigation

Based on the DFT calculations, electronic structures and reaction pathways of $Mo_2C$ (101) slab model with 10% of Mo atoms substituted by Ir (model shown in Supplementary Fig. 20 and cell parameter shown in Supplementary Table 6) were studied to deeply understand the origin of the high HOR activity of atomically dispersed $Ir_{SA}$-$Mo_2C/C$ catalyst at the atomistic level. The Ir (111) slab and the typical carbon-based $Ir_{SA}$-$N_4$-C single-atom catalyst (model shown in Supplementary Fig. 21) were also studied for comparison. Bader charge analysis (Supplementary Fig. 22) suggests the electron transfer from surface Mo to the adjacent Ir in $Ir_{SA}$-$Mo_2C$, consistent with the XPS results.The partial density of states (PDOS) plots (Fig. 5a) show that the Ir-5$d$ orbitals of $Ir_{SA}$-$N_4$-C are very localized. By contrast, the Ir-5$d$ orbitals of the $Ir_{SA}$-$Mo_2C$ exhibit a large degree of delocalization, suggesting its

metal-like electronic structure of Ir atoms in $Ir_{SA}$-$Mo_2C$, which is similar to the 5$d$ bands of Ir (111). Such maintained metallicity of Ir in $Ir_{SA}$-$Mo_2C$ was benefited from the Pt-like electronic structure of the $Mo_2C$ substrate, which makes the dispersed Ir sites behave more like single atom alloy and thus brings high HOR activity similar to Ir (111). Actually, it is noteworthy that almost all reported high-efficient HOR catalysts rely on the metallic sites[48]. Furthermore, the total density of states (DOS, Supplementary Fig. 23) of $Ir_{SA}$-$Mo_2C$ exhibits a higher occupation (with higher carrier concentration) at the Fermi level compared with $Ir_{SA}$-$N_4$-C, suggesting a much better electron conductivity[49].

The adsorption energies of the HOR intermediates control the activity of $Ir_{SA}$-$Mo_2C$. Although the mechanism of alkaline HOR was still under debate, H and OH adsorptions were generally considered as the key step in the kinetics of alkaline hydrogen oxidation[50–52]. Figure 5b summarizes the calculated adsorption Gibbs free energy ($\Delta G$) of adsorbed H, OH. The $\Delta G_H$ value is considered as the primary descriptor for the HOR kinetics. The H adsorptions on the different sites of $Ir_{SA}$-$Mo_2C$ were evaluated (Supplementary Fig. 24 and Supplementary Table 7), and it was found that the Ir-Mo hollow sites offer the suitable $\Delta G_H$ of −0.25 eV. The $Ir_{SA}$-$Mo_2C$ illustrated slightly weaker H binding than Ir (111) surface ($\Delta G_H$ = − 0.27 eV), which was more optimized for the HOR process[53]. The slightly weakened H binding was indicated by the 14 mV of negatively shift of the $H_{upd}$ peak as well (Fig. 5c). When an Ir cluster was placed on the $Mo_2C$ (101) substrate ($Ir_{NC}$-$Mo_2C$, model shown in Supplementary Fig. 25), a stronger H adsorption with $\Delta G_H$ = − 0.40 eV was found. Pure $IrO_2$ (110) ($\Delta G_H$ = − 0.64 eV) and $Mo_2C$ (101) ($\Delta G_H$ = − 0.57 eV) also exhibited the too strong H bindings (shown

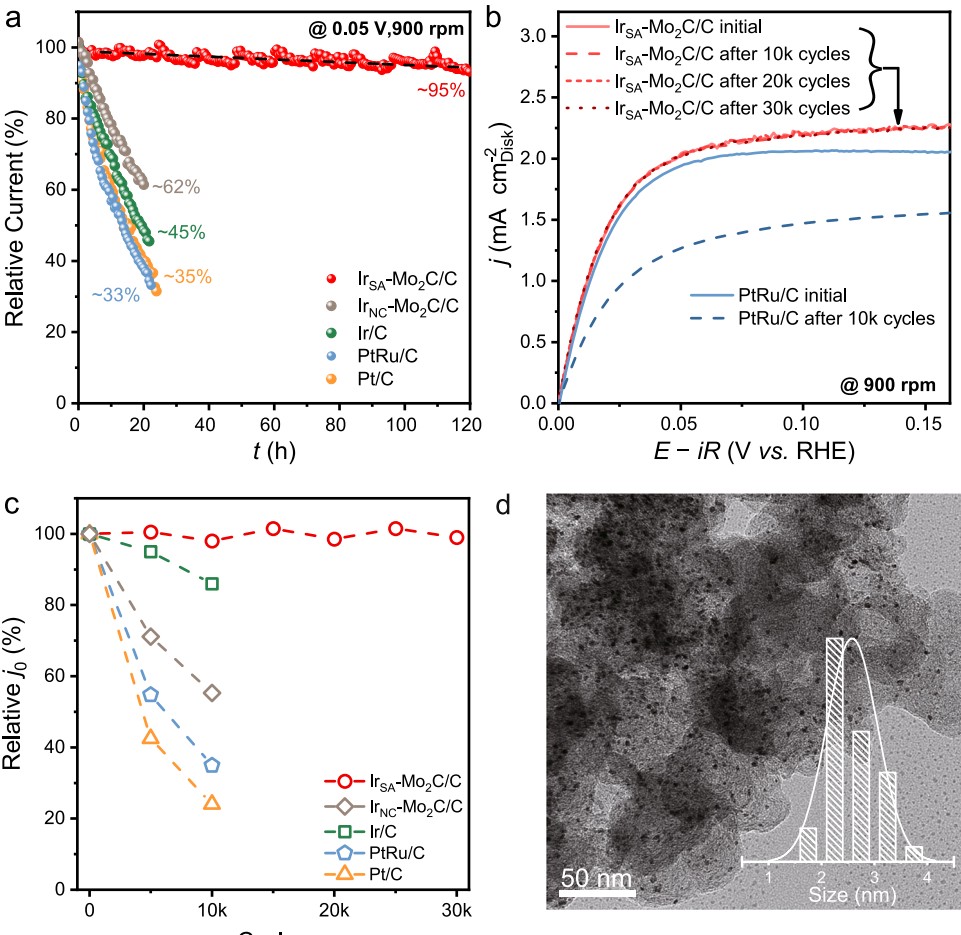

**Fig. 4 | Stability evaluation. a** The chronoamperometry at 50 mV of Ir$_{SA}$-Mo$_2$C/C, Ir$_{NC}$-Mo$_2$C/C, Ir/C, Pt/C and PtRu/C tested in H$_2$-saturated 0.1 M KOH with a rotation rate of 900 rpm. **b** polarization curves for Ir$_{SA}$-Mo$_2$C/C and PtRu/C before and after ADT. The scan rate was 2 mV s$^{-1}$ and the rotation rate was 900 rpm. The potentials were *iR* corrected and the *R* values for Ir$_{SA}$-Mo$_2$C/C and PtRu/C measurements were 37.0 ± 0.08 and 37.6 ± 0.10 Ω, respectively. **c** The relative exchange current density evolution in the ADT. **d** TEM image of the Ir$_{SA}$-Mo$_2$C/C after 30,000 cycles of ADT. The inset shows particle size distribution.

in Supplementary Fig. 26–27), leading to the poorer HOR activity, which were consistent to the experimental results. In addition, although Ir$_{SA}$-N$_4$-C demonstrated a stronger H binding ($\Delta G_H = -0.33$ eV) than Ir (111) surface, it did not remarkably deviate from the optimal value of H adsorption energy.

Besides $\Delta G_H$, the free energy of OH adsorption ($\Delta G_{OH}$) is considered as another important indicator for activity in alkaline HOR. Markovic et al. considered that the *OH combined with *H to generate water through the bifunctional mechanism, and thus accelerating the H desorption[11]. While Chen et al. proposed stronger H-bond network among interfacial water molecules, which was important for the H transfer[54]. Both possible mechanisms suggested that the enhanced OH adsorption could improve HOR. Here, the Ir$_{SA}$-Mo$_2$C, Ir and Ir$_{SA}$-N$_4$-C catalysts illustrated much different values of $\Delta G_{OH}$ (Fig. 5b). Based on the screening of the different sites of Ir$_{SA}$-Mo$_2$C, it was found that the Mo sites were more favored to adsorb OH (Supplementary Fig. 28 and Supplementary Table 6). Compared with Ir (111) surface ($\Delta G_{OH} = 0.66$ eV), Ir$_{SA}$-Mo$_2$C ($\Delta G_{OH} = 0.01$ eV) showed dramatic enhancement of OH adsorption, while Ir$_{SA}$-N$_4$-C ($\Delta G_{OH} = 1.02$ eV) possessed an increased $\Delta G_{OH}$ value. The enhanced OH adsorption of Ir$_{SA}$-Mo$_2$C could be concluded by the lower onset potential of Ir$_{SA}$-Mo$_2$C/C than Ir/C in the CO-stripping voltammetry (Fig. 5d). The much weaker OH adsorption of the Ir$_{SA}$-N$_4$-C catalyst should be the origin of its poor HOR performance.

Therefore, the calculated adsorption-free energies suggest the high HOR performance of the Ir$_{SA}$-Mo$_2$C/C can be attributed to its optimized H binding and enhanced OH binding. The Gibbs free energy profiles of the HOR process on each catalyst are displayed in Fig. 5e. Due to the weak OH adsorption, the potential-determining steps on Ir (111) and Ir$_{SA}$-N$_4$-C were both the OH$^-$ adsorption step (i.e., *H + * + OH$^-$ →*H + *OH + e$^-$), and the energy barriers were 0.70 and 1.86 eV, respectively. Four reaction paths were considered on Ir$_{SA}$-Mo$_2$C through different adsorption sites (Supplementary Fig. 29). Because of the enhanced OH adsorption on Mo sites, all the paths exhibited lower energy barrier for OH$^-$ adsorption step, giving the potential determining step of water generation step (i.e., *H + *OH → *H$_2$O) on Ir$_{SA}$-Mo$_2$C. The favorite path showed a lower energy barrier of 0.20 eV, indicating its fastest HOR kinetics. The Gibbs free energy profiles for Ir$_{NC}$-Mo$_2$C, Mo$_2$C (101) and IrO$_2$ (110) were calculated as well (Supplementary Fig. 25–27), which showed higher energy barriers of 0.80, 1.57, and 1.03 eV, respectively.

In summary, we have developed an atomically dispersed Ir (i.e., Ir$_{SA}$-Mo$_2$C/C) as a highly efficient and stable HOR catalyst in alkaline electrolyte to achieve high HEMFC performance using low anode PGM loading. The unique Mo$_2$C substrate properly adjusts the Ir sites to achieve optimized H binding and enhanced OH binding, and thus significantly promotes the HOR kinetics on Ir$_{SA}$-Mo$_2$C/C. The Ir$_{SA}$-Mo$_2$C/C exhibits a high mass activity of 17.9 A mg$_{Ir}^{-1}$ at 0.05 V,

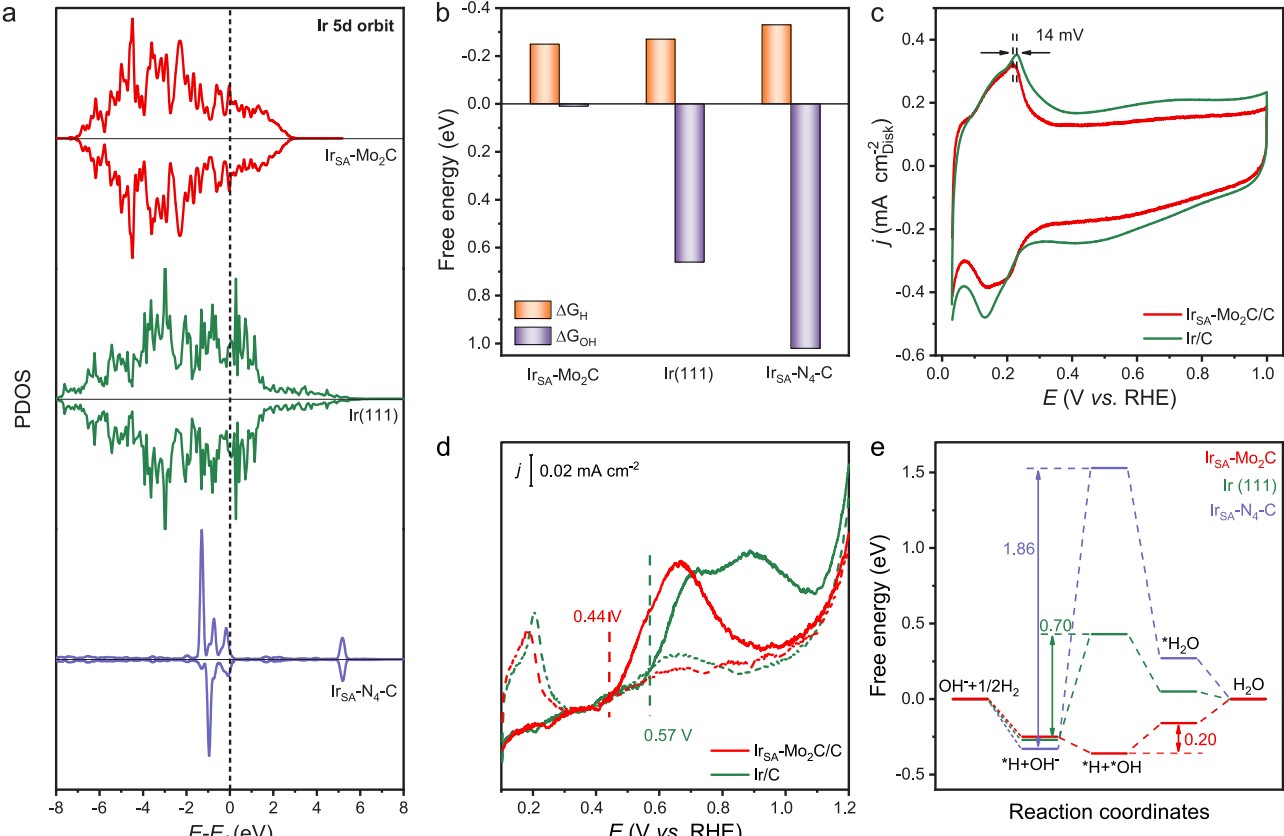

**Fig. 5 | Mechanism investigation. a** The PDOS diagram for $d$ orbital of metals in $Ir_{SA}$-$Mo_2C$, Ir (111) and $Ir_{SA}$-$N_4$-C. **b** The calculated adsorption Gibbs free energy of H and OH. **c** CV curves of Ir-$Mo_2C$/C and Ir/C obtained in Ar-saturated 0.1 M KOH without $iR$-correction. The scan rate is 20 mV s$^{-1}$. **d** CO-stripping curves of $Ir_{SA}$-$Mo_2C$/C and Ir/C obtained in 0.1 M KOH without $iR$-correction. The scan rate was 10 mV s$^{-1}$. The solid lines are CO-stripping voltammetry. The dash lines are CV curves after the CO-stripping. **e** The Gibbs free energy diagrams of HOR on the $Ir_{SA}$-$Mo_2C$, Ir (111) and $Ir_{SA}$-$N_4$-C.

surpassing the state-of-the-art PtRu/C by 4.3 times. The high stability of the $Mo_2C$ substrate also promotes the durability of the $Ir_{SA}$-$Mo_2C$/C, exhibiting a decay rate 72 times smaller than the PtRu/C. This work highlights that carbides can serve as an excellent substrate to host the atomically dispersed PGMs, and this type catalyst is promising in reducing the PGM requirement in electrocatalysis.

## Methods

### Materials
MoCl$_5$ (99.99%), Ethanol (99.5%), HNO$_3$ (65%) and KOH (99.999%) was purchased from Aladdin. IrCl$_3$·xH$_2$O (99.9%) was obtained from Sigma Aldrich. Ketjen black EC-300J was purchased from Lion Corporation. Nafion solution, Pt/C (20%, Johnson Matthey), PtRu/C (40% Pt and 20% Ru, Johnson Matthey), IrO$_2$ and gas diffusion layers (SGL 29BC) were obtained from Suzhou Sinero Technology Co., Ltd. The hydroxide exchange membrane (PiperION™ A15) and the ionomer were bought from Versogen. Ultrapure water (18.2 MΩ·cm, Mili-Q, Merck) was used in the synthesis of catalysts and electrochemical measurements.

### Syntheses of the catalysts
**Synthesis of the oxidized-carbon black.** 1 g of Ketjen black (EC-300J) was dispersed in 100 mL of 65% HNO$_3$ solution through ultrasonication for 15 min. The mixture was then heated to 85 °C with vigorous stirring at refluxed state for 1 h. Then it was centrifuged and dispersed in water, followed by filtering and washing with plenty of water. Finally, the filter cake was dried by lyophilization.

**Synthesis of the $Ir_{SA}$-$Mo_2C$/C catalyst.** Firstly, 200 mg of the as-prepared oxidized-carbon black and 380 mg of MoCl$_5$ were dispersed

in 30 mL of ethanol through ultrasonication for 15 min, followed by adding 5 mL of water. Then, the mixture was heat to 80 °C with vigorous stirring for 8 h. The product was centrifuged and dispersed in water, followed by filtering and washing with plenty of water. The MoO$_x$/C was obtained by lyophilization. Secondly, 0.03 mM of IrCl$_3$·xH$_2$O (12 mg) and 109 mg of the obtained MoO$_x$/C were dispersed in 50 mL of water by ultrasonication for 1 h. The solvent was then evaporated using rotary evaporator at 50 °C. The residue was collected and dried overnight. Then, it was grinded and calcined at 900 °C under N$_2$ atmosphere for 1 h to obtain $Ir_{SA}$-$Mo_2C$/C catalyst. The Ir loading in the catalyst was ca. 3.1% determined by ICP-OES.

**Synthesis of the $Ir_{NC}$-$Mo_2C$/C catalyst.** $Ir_{NC}$-$Mo_2C$/C catalyst was prepared by the same method as $Ir_{SA}$-$Mo_2C$/C catalyst, except using 3 times the amounts of IrCl$_3$·xH$_2$O.

**Synthesis of the Ir/C catalyst.** 0.03 mM of IrCl$_3$·xH$_2$O (12 mg) and 109 mg of oxidized-carbon black were dispersed in 50 mL of water by ultrasonication for 1 h. The solvent was then evaporated using rotary evaporator at 50 °C. The residue was collected and dried overnight. Then, it was grinded and calcined at 900 °C under N$_2$ atmosphere for 1 h to obtain Ir/C catalyst.

**Synthesis of the $Ir_{SA}$-$N_4$-C catalyst.** The synthesis of $Ir_{SA}$-$N_4$-C was followed by the procedure reported by Li et al.[42]. Typically, 2 mmol of Zn(NO$_3$)$_2$·6H$_2$O (594 mg) and 0.2 mmol of Ir(acac)$_3$ (98 mg) were simultaneously dissolved in 7.5 mL of methanol to form a uniform solution. Another solution was made by dissolving 8 mmol of 2-methyl imidazole (656 mg) to 15 mL of methanol. After rapidly mixing the two

solutions and vigorously stirring for 5 min, the mixed solution was transferred into a 50 mL Telfon-lined stainless-steel autoclave. The sealed vessel was heated at 120 °C for 4 h before it was cooled to room temperature. Then it was separated by centrifugation, washed with methanol for four times, and dried under vacuum for 8 h. Finally, it was put into a tube furnace and pyrolyzed at 900 °C for 3 h under Ar atmosphere to obtain $Ir_{SA}$-$N_4$-C catalyst.

## Characterizations

Powder X-ray powder diffraction (PXRD) patterns were recorded on a Rigaku D/Max 2500 VB2 + /PC X-ray powder diffractometer equipped with Cu $K_\alpha$ radiation (λ = 0.154 nm). Transmission electron microscopy (TEM) images were obtained on a HITACHI HT7700 transmission electron microscope operating at 100 kV. Aberration-corrected high-angle annular dark-field scanning transmission electron microscopy (AC-HAADF-STEM) and energy dispersive X-ray spectrometry (EDS) elemental mapping were performed using FEI TITAN 80-300 operating at 300 kV. The X-ray photoelectron spectra (XPS) were measured using a Thermo Fisher ESCALAB 250Xi XPS system with a monochromatic Al $K_\alpha$ X-ray source. All binding energies were calibrated to the C 1 s peak (284.8 eV). Inductively coupled plasma optical emission spectroscopy (ICP-OES, Optima 7300 DV, Perkin Elmer) was used to determine the metal contents.

## XAS measurements

X-ray absorption spectroscopy (XAS) at the Ir $L_3$-edge (11215 eV) were measured at 1W1B station in Beijing Synchrotron Radiation Facility (BSRF, operated at 2.5 GeV with a maximum current of 250 mA). The XAS data of the samples were collected at room temperature in fluorescence excitation mode using a Lytle detector. All the samples were mixed with graphite and ground uniformly and then pressed into a 10 mm plate with a thickness of 1 mm.

The EXAFS data were processed by Athena and Artemis in the IFEFFIT packages. Firstly, the EXAFS spectra were obtained by Fourier transforming the $\chi(k)$ data with Hanning windows (d$k$ = 0.5 Å$^{-1}$) to separate the contributions from different coordination shells. Then, quantitative structural parameters around the Ru atom were obtained by fitting with the following EXAFS equation using the Artemis module.

$$\chi(k) = \sum_j \frac{N_j S_0^2 F_j(k)}{k R_j^2} \exp\left[-2k^2\sigma_j^2\right] \exp\left[-\frac{2R_j}{\lambda(k)}\right] \sin\left[2kR_j + \phi_j(k)\right] \quad (1)$$

where $S_0^2$ is the amplitude reduction factor, $F_j(k)$ is the effective curved-wave backscattering amplitude, $N_j$ is the number of neighbor atoms from different coordination shells, $R_j$ is the distance between the core atom and neighbor atoms, $\lambda(k)$ is the mean free path in Å, $\phi_j(k)$ is the phase shift (including the phase shift for each shell and the total central atom phase shift), $\sigma_j^2$ is the Debye-Waller parameter of the different atomic shells. The functions $F_j(k)$, $\lambda(k)$ and $\phi_j(k)$ were calculated by the ab initio code FEFF 8.2. During the fitting process, $S_0^2$ was fixed, while R, $\sigma^2$ and the edge-energy shift $\Delta E_0$ were allowed to run freely.

## Electrochemical measurements

The electrochemical measurements were operated at room temperature in a standard three-electrode electrochemical system controlled by a potentialstat (V3, Princeton Applied Research). A homemade glass electrochemical cell equipped with three-electrode assembly was used. The cell and glassware were cleaned by 3:1 mixture of concentrated sulfuric acid and hydrogen peroxide (30%), rinsed thoroughly and boiled with ultrapure water before measurements. A catalyst thin film on glassy carbon (GC) rotating disk electrode RDE (5 mm in diameter, Pine Research Instrumentation) was used as the working electrode. The surface of GC was polished with 0.05 mm $Al_2O_3$ powder and cleaned with water under ultrasonic. The catalyst ink was made by ultrasonically dispersing 1 mg of as-prepared $Ir_{SA}$-$Mo_2C$/C

catalyst in a mixed solution containing 250 μL of distilled water, 750 μL of ethanol, and 5 μL of 5 wt % Nafion. Then 20 μL of the catalyst ink was transferred onto the GC substrate to yield a thin film electrode. The Ir loading was ca. 3.2 μg$_{Ir}$ cm$^{-2}$ according to the result of ICP-OES. The reversible hydrogen electrode (RHE, Phychemi Co., Ltd.) and a Pt wire (99.999%, Gaoss Union Co., Ltd.) were used as the reference and counter electrodes, respectively. The fresh 0.1 M KOH solution with the pH of ca. 12.8 ± 0.1, prepared by dissolving 2.81 g of KOH in 500 mL of water, was used as the electrolyte for all RDE measurements.

Cyclic voltammetry (CV) was carried out to analyze the electrochemical properties of the catalysts. The recorded potentials were iR-corrected. The solution resistance (R) was measured using AC-impendence from 200 kHz to 100 mHz with a voltage perturbation of 5 mV. The HOR activity was calculated using a positive-going direction polarization curves obtained in a $H_2$-saturated 0.1 M KOH solution with the rotation rate of 1600 rpm and the scanning rate of 5 mV s$^{-1}$. For the stability test, the catalyst loading was controlled at ca. 7 μg$_{Ir}$ cm$^{-2}$. A homemade fluorinated ethylene propylene electrochemical cell equipped was used. Two techniques were used to evaluated the stability. The rotation rate was change to 900 rpm to reduce the mechanical detachment of catalyst from the surface of RDE. Firstly, the chronoamperpmentry at a constant potential of 0.05 V was used. Secondly, the CV cycling was used for the accelerated durability test (ADT) by scanning the potential between 0 and 0.5 V with the scanning rate of 100 mV s$^{-1}$. To eliminate the influence of $CO_2$ from air, the electrolyte was refreshed each 50 h.

## Evaluation of activity and ECSA

The HOR/HER activity was represented by the exchange current density ($j_0$), which was obtained by fitting kinetic current ($i_k$) with the Butler-Volmer equation:

$$i_k = j_0 A_s \left[\exp\left(\frac{\alpha_a F\eta}{RT}\right) - \exp\left(\frac{-\alpha_c F\eta}{RT}\right)\right] \quad (2)$$

where $\alpha_a$ and $\alpha_c$ are the transfer coefficients for HOR and HER, respectively, with $\alpha_a + \alpha_c = 1$ (Tafel-Volmer pathway). $A_s$ is the electrochemical surface area (ECSA), and $\eta$ is the overpotential. $i_k$ is calculated by the Koutecky-Levich equation:

$$\frac{1}{i} = \frac{1}{i_k} + \frac{1}{i_d} \quad (3)$$

where $i$ is the measured current, $i_k$ is the kinetic current, and $i_d$ is the diffusion limited current. $i_d$ is defined from the following equation:

$$i_d = i_l \left[1 - \exp\left(-\frac{2F\eta}{RT}\right)\right] \quad (4)$$

where $\eta$ is the overpotential and $i_l$ is the maximum current obtained from polarization curves.

The ECSA was evaluated by CO-stripping voltammetry. A monolayer of CO was firstly adsorbed onto the catalyst's surface by holding at 0.1 V for 10 min. The electrolyte was then flushed with Ar for 10 min to remove dissolved CO completely. Then, the adsorbed CO was stripped by scanning between 0.1 and 1.2 V at a scan rate of 10 mV·s$^{-1}$.

## HEMFCs fabrications and tests

The synthesized $Ir_{SA}$-$Mo_2C$/C or the control catalysts were used as the anode catalyst with the loading of 0.05 mg$_{PGM}$ cm$^{-2}$. Pt/C (HiSpec 4000, 40 wt% Pt, Alfa Aesar) was used as the cathode catalyst with the loading of 0.4 mg$_{Pt}$ cm$^{-2}$. The catalyst ink was sprayed onto both sides of PiperION™ A15 membrane (17 μm) to fabricate a catalyst coated membrane (CCM) with the electrode area of 5 cm$^{-2}$. All CCMs were immersed into 3 M NaOH solution overnight and then rinsed

thoroughly with deionized water to remove all excess NaOH. The rinsed CCM was assembled with a fluorinated ethylene propylene (FEP) gasket, a GDL (SGL 29 BC), a graphite bipolar plate with 5 cm$^2$ flow field and a metal current collector for each side to complete the full HEMFC. Fuel cell test system (Scribner 850 g) equipped with a backpressure module was used to evaluate the performance of the HEMFCs. The operating temperature was set as 95 °C. The $H_2$ and $O_2$ were humidified at 92 °C and 95 °C, respectively. The flow rates for both $H_2$ and $O_2$ were 1000 sccm. The back-pressure was set at 250 kPa.

## DFT calculations

First-principles computations were performed using the projector augmented wave method (PAW) as implemented in the Vienna ab-initio simulation package (VASP). The generalized gradient approximation in the form of Perdew-Burke- Ernzerhof (PBE) for the exchange-correction potential and a cutoff energy of 500 eV for the plane-wave basis were adopted. For all calculations, spin polarization was adopted. To consider long-range van der Waals (vdW) interactions, Grimme's DFT-D3 method was employed. Atomic structures were fully released with converging tolerance for forces on all atoms less than 0.02 eV Å$^{-1}$, and the energy convergence criterion was set to $10^{-5}$ eV. 2×2, 5×5, 2×3, 2×3 and 3×2 unit cells of Ir (111), $Ir_{SA}$-$N_4$-C, $Ir_{SA}$-$Mo_2$C,$Ir_{NC}$-$Mo_2$C and $IrO_2$ (110) with 4 layers thickness was employed with 15 Å vacuum space in the vacuum-direction to avoid image interactions. The bottom two layers were fixed while the top two layers were relaxed during geometry optimization. 3×3×1, 2×1×2, 3×3×1, 3×3×1 and 3×2×1 Monkhorst–Pack k-meshes were used to sample the Brillouin zone for geometry relaxation. The computational hydrogen electrode model proposed by Nørskov et al.[55]. Was used to evaluate the activity of the catalyst. The Gibbs free energy change, $\Delta G$, of each elementary step was obtained based on the following equation, in which the contributions of zero-point energy ($\Delta ZPE$) and entropy ($T\Delta S$) changes were considered:

$$\Delta G = \Delta E + \Delta ZPE - T\Delta S - eU \qquad (5)$$

Where $\Delta E$ was the calculated energy change of the intermediates, such as *H, *OH and *$H_2$O, $U$ was the applied electrode potential.

## Data availability

The data generated in this study are provided in the Source Data file. Source data are provided with this paper.

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

## Acknowledgements

This work was financially supported by the National Key Research and Development Program of China (2019YFA0210300, ZZ), the National Natural Science Foundation of China (22379004 ZZ, 22101016 WZ, 22288102 HL), Beijing Natural Science Foundation (Z210016, ZZ), and Fundamental Research Funds for the Central Universities (buctrc201916, ZZ). The authors appreciate the 1W2B in the Beijing Synchrotron Radiation Facility (BSRF) for help with XAFS experiments.

## Author contributions

Z.Z. supervised the project. Z.Z. and J.F. conceived the idea. J.F. and Hai.W synthesized and characterized the catalysts. J.P. and Z.X conducted the XAS experiments. J.F. and Hai.W carried out the electrochemical measurements. J.F. and X.W. assembled the HEMFCs. Q.D. and H.L. performed the DFT calculations. Hao.W., C.C., and W.Z. helped with data interpretation. J.F., H.L., Y.Y., and Z.Z. wrote the manuscript. All authors discussed the results and assisted with the manuscript preparation.

## Competing interests

The authors declare no competing interests.
