## [Peer Review File · Nature Communications]

Reviewers' Comments:

Reviewer #1:

Remarks to the Author:

In this manuscript, Fang et al. reported the Ir on Mo₂C as highly active catalyst for HOR catalyst in alkaline medium. Different from the commonly used carbon-based single atom Ir electrocatalyst, the reported Ir-Mo₂C showed high HOR performance, which is better than the commercial Pt/C. The authors attributed this high activity to the Mo₂C substrate. A high fuel cell performance was achieved, with reduced anode PGM loadings to the fuel cell using Pt/C or PtRu/C. The results are interesting and provide new knowledge to the development of low-cost and effective HOR catalysts for fuel cells. Overall, this work is well conducted, and the manuscript can be considered for publication in the Nature Communications after minor revisions.

1. How about the Ir and Mo contents from EDS?
2. The peaks in the XPS of Ir/C are asymmetric, demonstrating that multiple species co-existed in the sample. Thus, deconvolution of the peaks is required.
3. The ECSA of the catalysts was measured through the CO stripping method. What is the specific charge used in the ECSA calculation? Because the Ir-Mo₂C is single atom catalyst, what is the definition of ECSA for Ir-Mo₂C?
4. The fuel cell performance under H₂-air condition should be provided as well, because the H₂-O₂ condition rarely exists in real applications.
5. Page 11, "while Ir-N₄-C (Δ GOH = 1.02 eV) possessed a decrease Δ GOH value". The Δ GOH of Ir-N₄-C is largest compared with other catalysts.

Reviewer #2:

Remarks to the Author:

In this manuscript, Ir single-atom supported on Mo₂C has been reported with excellent activity and stability for alkaline HOR. This catalyst not only displayed high HOR performance in RDE tests, it also showed high AEMFC performance, indicating by the high power density of AEMFC achieved with a low anode PGM loading. The mechanism of the Ir/Mo₂C catalyzed HOR process was discussed, and the advantages of the Mo₂C substrate was illustrated. This reviewer recommends the publication of this manuscript in Nature Communications after minor revision listed below.

1. The absorption edges overlapped in Figure 2c, and it is difficult to see the edge position of Ir-Mo₂C. The EXAFS fitting results of Ir foil and IrO₂ should also be provided as reference.
2. The authors provide the Ir XPS results. Because the authors emphasized the role of Mo₂C substrate, it is recommended to provide the Mo XPS spectra as well.
3. Why different PGM loadings used in the RDE test shown in Figure 3a? In figure 3f, the "Peak powder density" should be "Peak power density".
4. For AEMFCs, they generally work at 80 C. In this manuscript, a high working temperature of 95 C was used. The authors should give explanations on the choice of the cell temperature.
5. The Gibbs free energy diagrams of HOR on the Ir-Mo₂C, Ir and Ir-N₄-C were provided in Figure 5e. However, for the controls of IrO₂ and Mo₂C, the authors only gave their hydrogen binding energies. How about the Gibbs free energy diagrams for IrO₂ and Mo₂C? Do they show high energy barrier for the HOR processes?

Reviewer #3:

Remarks to the Author:

This work has prepared a carbon-supported atomically dispersed Ir-Mo₂C catalyst. The authors find that this catalyst exhibits high activity and stability for HOR compared to Pt/C and PtRu/C catalysts. They speculate that the Mo₂C substrate plays a crucial role in improving the HOR activity and stability of the Ir-based catalysts. However, the exact reason behind this improvement lacks discussion and is still unclear. Then, this manuscript could be considered for publication in "Nature Communication" if the following suggested issue is addressed successfully.

1. The introduction of the research does not provide sufficient reasons for choosing Ir as the catalyst and Mo₂C as the support. Additionally, the analysis and summary of the existing research

on HOR catalysts, especially Ir-based catalysts, are lacking. It is recommended to add these summaries to the introduction part.

2. In the introduction, the authors mentioned the corrosion of carbon supports in alkaline media, but they still used carbon as the support to prepare the Ir-Mo₂C/C catalysts. Please provide the basis for this choice. It is suggested to add a summary of current research on the selection of non-carbon supports, such as oxides, carbides, and nitrides, to act as HOR supports.

3. Experimental characterization and electrochemical tests have confirmed that the Ir-Mo₂C catalyst, with its atomically dispersed structure, exhibits excellent activity and stability for HOR. However, there is a lack of analysis regarding the contribution of this monoatomic dispersion to the activity and stability. To investigate this further, it is recommended to prepare cluster-dispersed Ir/Mo₂C catalysts and examine the role of atomic dispersion structure on activity and stability, as well as the specific effect of Mo₂C supports on activity and stability.

4. The Ir-N₄C monodisperse catalysts were not prepared and tested in the experiment. Therefore, it is not necessary to use DFT calculations to compare IrMo₂C and IrN₄C models. It is suggested to compare the activity of atomically dispersed and cluster dispersed Ir-Mo₂C catalysts using theoretical calculation.

5. Fig. 5 shows that the Gibbs free energy of H on the Ir site of Ir-Mo₂C is much larger than that of platinum (Pt) (about -0.15 eV). However, Ir-Mo₂C has higher HOR activity than Pt/C. H binding energy cannot explain the enhanced activity. How to understand the improved activity? In addition, it is unclear whether the active site for HOR in Ir-Mo₂C is Ir or Mo. In order to confirm the reaction mechanism and enhanced catalytic activity and stability, the optimal active sites for H and OH adsorption, the co-adsorption of H and OH, and the whole HOR mechanism need to be calculated and determined.

6. Please provide the side view of the corresponding models, and the cell structure and cell parameters of both Mo₂C and Ir-Mo₂C models. To validate the calculation model and reaction mechanism, it is recommended to provide the electronic structure of the corresponding crystal surface (such as Bader charge, pdos, etc.) for comparison with the corresponding valence data in experiments.

7. Can the monoatomic dispersed Ir-Mo₂C/C catalyst resist CO poisoning?

RESPONSE TO THE REVIEWERS' COMMENTS

We sincerely thank all the reviewers for careful attention to our manuscript and valuable, constructive comments. The comments from reviewers are copied in *italics* and our point-by-point responses to each comment are given in **blue**.

Reviewer #1:

General Comments: *In this manuscript, Fang et al. reported the Ir on Mo₂C as highly active catalyst for HOR catalyst in alkaline medium. Different from the commonly used carbon-based single atom Ir electrocatalyst, the reported Ir-Mo₂C showed high HOR performance, which is better than the commercial Pt/C. The authors attributed this high activity to the Mo₂C substrate. A high fuel cell performance was achieved, with reduced anode PGM loadings to the fuel cell using Pt/C or PtRu/C. The results are interesting and provide new knowledge to the development of low-cost and effective HOR catalysts for fuel cells. Overall, this work is well conducted, and the manuscript can be considered for publication in the Nature Communications after minor revisions.*

Response: Thanks for your remarks and kind recommendation for our work. All your comments are highly valuable for us to improve the manuscript. We addressed the comments point-by-point and made the corresponding changes accordingly in the revised manuscript and SI.

Comment 1. *How about the Ir and Mo contents from EDS?*

Response: We thank the reviewer for the valuable question. The EDS spectra of Ir_{SA}-Mo₂C/C is shown in Figure R1. The Ir/Mo molar ratio is 1:11, which is close to the result of 1:10 obtained from ICP-OES.

Figure R1. EDS spectra of Ir_{SA}-Mo₂C/C.

Revision: Page 4, “The Ir/Mo molar ratio in Ir_{SA}-Mo₂C/C was ca. 1:11 from EDS spectra (Supplementary Fig. 3), which was consistent with the ratio of ca. 1:10 tested by inductively coupled plasma optical emission spectra (ICP-OES).”

Comment 2. *The peaks in the XPS of Ir/C are asymmetric, demonstrating that multiple species co-existed in the sample. Thus, deconvolution of the peaks is required.*

Response: Thanks for your valuable suggestion. We have deconvoluted the Ir 4f XPS spectra and the results are shown in Figure R2. For Ir/C, the 4f_{7/2} peak could be deconvoluted into three peaks: Ir⁰ at 61.5 eV, Ir⁴⁺ at 62.4 eV and its satellite peak at 64.0 eV. Similar deconvolution results were adopted for the 4f_{5/2} peak. The existence of the Ir⁴⁺ species is likely from the surface oxidation. For IrO₂, only the Ir⁴⁺ species and the satellite peak exist. For Ir_{SA}-Mo₂C/C, the peaks are located in between of those for Ir⁰ and Ir⁴⁺.

Figure R2. High resolution Ir 4f XPS spectra of Ir_{SA}-Mo₂C/C, Ir/C and IrO₂.

Revision: Page 5, “The 4f_{7/2} peak of Ir/C could be deconvoluted into three peaks: Ir⁰ at 61.5 eV, Ir⁴⁺ at 62.4 eV and its satellite peak at 64.0 eV. The existence of the Ir⁴⁺ species is likely from the surface oxidation of nanoparticles. For IrO₂, only the Ir⁴⁺ species and satellite peak were observed. The peak of Ir_{SA}-Mo₂C/C was located in between of those for Ir⁰ and Ir⁴⁺, indicating its medium oxidation state.”

Comment 3. *The ECSA of the catalysts was measured through the CO stripping method. What is the specific charge used in the ECSA calculation? Because the Ir-Mo₂C is single atom catalyst, what is the definition of ECSA for Ir-Mo₂C?*

Response: We thank the reviewer for the valuable question. The specific charge we used for the ECSA calculation of the Ir_{SA}-Mo₂C/C sample was 420 μC cm⁻², which was used for metallic Ir (*Nat. Commun.* 2023, 14, 5402). It is difficult to define the ECSA for single atom catalysts. We used the ECSA normalized activity to compare the intrinsic activity of the catalysts, which is correlated to the activity of each site. Thus, we used the same specific charge to calculate the ECSAs for all Ir-based catalyst, to ensure the ECSA normalized activity can represent the intrinsic activity of the Ir sites. We have clarified it in the revised manuscript.

Revision: Page 7, “The ECSA of the catalysts are determined by CO stripping voltammetry, and the same specific charge of 420 μC cm⁻² was used for all the catalysts”.

Comment 4. *The fuel cell performance under H₂-air condition should be provided as well, because the H₂-O₂ condition rarely exists in real applications.*

Response: Thanks for your valuable suggestion. The HEMFC with the same catalyst loadings was tested under H₂-air condition and shown in Figure R3. The Ir_{SA}-Mo₂C/C-based HEMFC delivered a PPD of 0.9 W cm⁻², and reached the current density of 1.11 A cm⁻² at 0.65 V. This figure is

provided in the revised SI.

Figure R3. H₂/air (CO₂-free) HEMFC performance using Ir_{SA}-Mo₂C/C as anode catalyst.

Revision: Page 8, “The Ir_{SA}-Mo₂C/C HEMFC also showed high performance under H₂/air(CO₂-free) condition, which delivered a PPD of 0.90 W cm⁻² and reached a current density of 1.11 A cm⁻² at 0.65 V (Supplementary Fig. 12).”

Comment 5. Page 11, “while Ir-N₄-C ($\Delta G_{OH} = 1.02$ eV) possessed a decrease ΔG_{OH} value”. The ΔG_{OH} of Ir-N₄-C is largest compared with other catalysts.

Response: We apologize for the error. We have corrected it as “while Ir_{SA}-N₄-C ($\Delta G_{OH} = 1.02$ eV) possessed an increased ΔG_{OH} value.” in the revised manuscript.

Reviewer #2:

General Comments: *In this manuscript, Ir single-atom supported on Mo₂C has been reported with excellent activity and stability for alkaline HOR. This catalyst not only displayed high HOR performance in RDE tests, it also showed high AEMFC performance, indicating by the high power density of AEMFC achieved with a low anode PGM loading. The mechanism of the Ir/Mo₂C catalyzed HOR process was discussed, and the advantages of the Mo₂C substrate was illustrated. This reviewer recommends the publication of this manuscript in Nature Communications after minor revision listed below.*

Response: We thank the reviewer for the kind comments of our work. All your comments are highly valuable for us to improve the manuscript. We addressed the comments point-by-point and made the corresponding changes accordingly in the revised manuscript and SI.

Comment 1. *The absorption edges overlapped in Figure 2c, and it is difficult to see the edge position of Ir-Mo₂C. The EXAFS fitting results of Ir foil and IrO₂ should also be provided as reference.*

Response: Thanks for your valuable suggestion. The enlarged plot of edge position is shown in the Figure R4b. The Ir absorption edge of Ir_{SA}-Mo₂C/C is located in between those of Ir foil and IrO₂. However, the edge position of Ir L₃ is insensitive to the valence state, with Ir foil just showing a slight left-shift compared with IrO₂. The white line intensity is more sensitive to the valence state for Ir (*Nat. Catal.* 2023, 6, 916). Ir_{SA}-Mo₂C/C clearly shows the white line intensity between Ir foil and IrO₂. We have revised the manuscript to clarify it.

Figure R4. (a) Ir L₃-edge XANES spectra of Ir_{SA}-Mo₂C/C and the reference samples of IrO₂ and Ir foil. (b) The enlarged plot of edge position.

The EXAFS fitting has also been carried out for the reference samples. The results are shown in Figure R5 and Table R1, and provided in the revised SI.

Figure R5. FT-EXAFS fitting curves of Ir foil at (a) R space, (b) k space and (c) q space. FT-EXAFS fitting curves of IrO_2 at (d) R space, (e) k space and (f) q space.

Table R1. FT-EXAFS fitting results of $\text{Ir}_{\text{SA}}\text{-Mo}_2\text{C/C}$, Ir foil and IrO_2 .

Sample	Scattering pair	$R(\text{\AA})$	N	$\sigma^2 (10^{-3}\text{\AA}^2)$	ΔE_0 (eV)	R factor
$\text{Ir}_{\text{SA}}\text{-Mo}_2\text{C/C}$	Ir-C	2.10	3.12	3.89	11.5	0.008
	Ir-Mo	2.70	1.40	0.52		
Ir foil	Ir-Ir	2.72	12.0	2.41	9.64	0.005
IrO_2	Ir-O	1.98	5.86	3.49	12.5	0.016

Revision: Page 5, “The Ir L_3 -edge X-ray absorption near-edge structure (XANES, Fig. 2a) of $\text{Ir}_{\text{SA}}\text{-Mo}_2\text{C/C}$ exhibited the white line intensity between that of Ir foil and IrO_2 , revealing that the valence state of Ir in $\text{Ir}_{\text{SA}}\text{-Mo}_2\text{C/C}$ was between that of Ir and IrO_2 .”

Page 5, “The fitting curves for Ir foil and IrO_2 are shown in Supplementary Fig. 6 and the fitting results are summarized in Supplementary Table 1.”

Comment 2. The authors provide the Ir XPS results. Because the authors emphasized the role of Mo_2C substrate, it is recommended to provide the Mo XPS spectra as well.

Response: We thank the reviewer for the valuable suggestion. The Mo 3d XPS spectra and the

corresponding deconvolution result of Ir_{SA}-Mo₂C/C and Mo₂C are shown in Figure R6. The peaks at 228.8, 229.8 and 232.7 eV are assigned to the 3d_{5/2} peak of Mo²⁺, Mo⁴⁺ and Mo⁶⁺ species, respectively.⁶ It is found that cooperating with Ir, the proportion of Mo⁶⁺ species increased, demonstrating the tuning effect.

Figure R6. High resolution Mo 3d XPS spectra of Ir_{SA}-Mo₂C/C and Mo₂C/C.

Revision: Page 5, “Supplementary Fig. 4 shows the Mo 3d XPS spectra of Ir_{SA}-Mo₂C/C and Mo₂C/C, indicating the proportion of Mo⁶⁺ species increased after introducing Ir.”

Comment 3. Why different PGM loadings used in the RDE test shown in Figure 3a? In Figure 3f, the “Peak powder density” should be “Peak power density”.

Response: We thank the reviewer for the valuable question and notification. We have redone the tests and used the same PGM loadings of 3.2 μg_{PGM} cm⁻² for all the samples. The results are shown in Figure R7. Similar specific and mass activity are obtained compared with the previous results using the PGM loadings of 5.0 μg_{PGM} cm⁻². And the results are updated in the revised manuscript.

The typo in the x-axis title in Figure 3f is corrected in the revised manuscript.

Figure R7. (a) Polarization curves of the catalysts in H₂-saturated 0.1 M KOH solution with the rotation rate of 1600 rpm and the scan rate of 5 mV s⁻¹. The PGM loading is 3.2 μg_{PGM} cm⁻² for all catalysts. (b) The corresponding Tafel plots. Inset shows the specific exchange current density ($j_{0,ECSA}$).

Comment 4. For AEMFCs, they generally work at 80 °C. In this manuscript, a high working

temperature of 95 °C was used. The authors should give explanations on the choice of the cell temperature.

Response: Thanks for your valuable question. In our work, we used the PiperION™-A15 membrane (purchased from Versogen™, reported in *Nat. Energy* 2019, 4, 392), which offers excellent stability and high performance with hydroxide conductivity of 193 mS cm⁻¹ at 95 °C, allowing the HEMFCs work at 95 °C. The high working temperature brings better performance. Thus, we chose to test the HEMFC performance at 95 °C. And we tested all the MEAs at the same temperature, so the comparison was fair.

Comment 5. The Gibbs free energy diagrams of HOR on the Ir-Mo₂C, Ir and Ir-N₄-C were provided in Figure 5e. However, for the controls of IrO₂ and Mo₂C, the authors only gave their hydrogen binding energies. How about the Gibbs free energy diagrams for IrO₂ and Mo₂C? Do they show high energy barrier for the HOR processes?

Response: Thanks for your valuable questions. We calculated the Gibbs free energy diagrams for Mo₂C (101) and IrO₂ (110) as shown in Figure R8. The potential-determining step on Mo₂C is the water generation step (i.e., *H + *OH → *H₂O) with a high energy barrier of ~1.57 eV. While the potential-determining step on IrO₂ is the water desorption step (i.e., *H₂O → H₂O) with an energy barrier of ~1.03 eV.

Figure R8. The Gibbs free energy diagrams of HOR on the (a) Mo₂C (101) and (b) IrO₂ (110).

Revision: Page 12, “The Gibbs free energy profiles for Ir_{NC}-Mo₂C, Mo₂C (101) and IrO₂ (110) were calculated as well (Supplementary Fig. 24-26), which showed higher energy barriers of 0.80, 1.57 and 1.03 eV, respectively.”

Reviewer #3:

General Comments: *This work has prepared a carbon-supported atomically dispersed Ir-Mo₂C catalyst. The authors find that this catalyst exhibits high activity and stability for HOR compared to Pt/C and PtRu/C catalysts. They speculate that the Mo₂C substrate plays a crucial role in improving the HOR activity and stability of the Ir-based catalysts. However, the exact reason behind this improvement lacks discussion and is still unclear. Then, this manuscript could be considered for publication in “Nature Communication” if the following suggested issue is addressed successfully.*

Response: Thanks for your great efforts in reviewing our manuscript. All your comments are highly valuable for us to improve the manuscript. We addressed the comments point-by-point and made the corresponding changes accordingly in the revised manuscript and SI.

Comment 1. *The introduction of the research does not provide sufficient reasons for choosing Ir as the catalyst and Mo₂C as the support. Additionally, the analysis and summary of the existing research on HOR catalysts, especially Ir-based catalysts, are lacking. It is recommended to add these summaries to the introduction part.*

Response: We thank the reviewer for the valuable comment.

Improvement of the HOR performance in alkaline condition is important for HEMFCs. The Pt based HOR catalysts have been widely used and the PtRu was used as the benchmark catalyst because the addition of Ru provides better OH adsorption. Ir has similar H binding to Pt, but with enhanced OH binding, and thus Ir has shown promising HOR activity in alkaline (*Nat. Chem.* 2013, 5, 300). Thus, we chose Ir and the research on Ir-based catalysts was summarized in the revised manuscript.

The Mo₂C shows Pt-like electronic structure, and it also has high stability and strong interaction with PGMs (*Joule* 2017, 1, 253). Thus, we chose Mo₂C as the substrate to obtain the benefit of strong interaction with Ir as well as the electronic tuning effect.

Revision: Page 2, “Ir, which demonstrates enhanced OH bindings than Pt, has been shown to be a promising candidate for HOR in alkaline. Ir alloy (e.g., IrNi, IrMo and IrRu) and heterostructures (e.g., Ir/MoS₂ and Ir/WO_x) have been reported with enhanced HOR activities, even comparable to the state-of-the-art PtRu/C catalysts.”

Page 3, “The Mo₂C shows Pt-like electronic structure, enabling the guest Ir to have a unique binding property with intermediates. Mo₂C also has high stability and strong interaction with PGMs, making Ir atoms atomically dispersed in the hexagonal Mo₂C matrix.”

Comment 2. *In the introduction, the authors mentioned the corrosion of carbon supports in alkaline media, but they still used carbon as the support to prepare the Ir-Mo₂C/C catalysts. Please provide the basis for this choice. It is suggested to add a summary of current research on the selection of non-carbon supports, such as oxides, carbides, and nitrides, to act as HOR supports.*

Response: We thank the reviewer for the valuable question. Carbon support was widely used in fuel cell catalysts, which provides high surface areas as well as excellent electron conductivity. However, the carbon supports suffered from corrosion, especially when in contact with PGMs, which catalyze the carbon corrosion. Alternative supports, such as oxides, carbides, and nitrides, have been reported. However, they generally have unsatisfied surface areas and electron conductivity, or they are

difficult to synthesize. And there is still no commercially available support other than carbon.

A compromise solution is to use a buffer to separate PGMs and the carbon supports. For example, Dekel et al. reported Pd/CeO₂-C with enhanced stability, by using CeO₂ as buffer substrate to load Pd nanoparticles, which could prevent the direct contact between Pd sites and the carbon support and thus mitigate the local carbon corrosion (*Angew. Chem. Int. Ed.* 2016, 55, 6004). We employed the same approach, and used Mo₂C as the buffer to avoid the direct interaction of Ir to the carbon support. Enhanced stability was achieved.

Revision: page 2, “Alternative supports, such as oxides, carbides, and nitrides, have been reported to improve the stability. For example, TiO₂-RuO₂ was used as the support for Pt nanoparticles, and enhanced fuel cell stability was achieved. Another method is to use a buffer to separate PGMs and the carbon substrates. For example, Dekel et. al. reported Pd/CeO₂-C with enhanced stability, by using CeO₂ as buffer substrate to load Pd nanoparticles, which could prevent the direct contact between Pd sites and the carbon support and thus mitigate the local carbon corrosion.”

Comment 3. *Experimental characterization and electrochemical tests have confirmed that the Ir-Mo₂C catalyst, with its atomically dispersed structure, exhibits excellent activity and stability for HOR. However, there is a lack of analysis regarding the contribution of this monoatomic dispersion to the activity and stability. To investigate this further, it is recommended to prepare cluster-dispersed Ir/Mo₂C catalysts and examine the role of atomic dispersion structure on activity and stability, as well as the specific effect of Mo₂C supports on activity and stability.*

Response: We thank the reviewer for the valuable suggestion. We prepared Ir nanocluster on Mo₂C supported on carbon (Ir_{NC}-Mo₂C/C) by increasing the Ir content. Figure R9 shows the XRD, HADDF-STEM and EDS-mapping images of Ir_{NC}-Mo₂C/C, which demonstrates the successful synthesis of Ir_{NC}-Mo₂C/C. XRD pattern (Figure R9a) shows the coexistence of crystallized Mo₂C and Ir. HADDF-STEM image (Figure R9b) shows the well dispersed nanoparticles with the average size of 3.7 nm. The high-resolution image (Figure R9c) illustrates two sets of lattices with spacing of 0.21 and 0.23 nm in one particle, which can be assigned to the (111) facet of Ir and (101) facet of Mo₂C, respectively. Furthermore, EDS-mapping results (Figure R9d) show the heterogeneously dispersion of Ir and Mo in this particle, suggesting the Ir nanocluster on Mo₂C.

Figure R9. Characterization of the Ir_{NC}-Mo₂C/C. (a) XRD pattern. (b-c) HAADF-STEM images. The inset shows the particle size distribution. (d) EDS-mapping of the particle shown in c.

Figure R10 shows the activity and stability of Ir_{NC}-Mo₂C/C. Ir_{NC}-Mo₂C/C exhibits an inferior HOR activity of $j_{0,s}$ of $1.2 \text{ mA cm}^{-2}_{\text{Ir}}$, much lower than that of Ir_{SA}-Mo₂C/C, demonstrating the benefit of the monoatomic dispersed Ir in Mo₂C. Ir_{NC}-Mo₂C/C also exhibits worse stability compared with Ir_{SA}-Mo₂C/C. Specifically, Ir_{NC}-Mo₂C/C maintained only 61% of the initial activity after continuous working for 20 h at 50 mV with the decay rate of $1.9\% \text{ h}^{-1}$ (Ir_{SA}-Mo₂C/C maintained 95% after 120 h with decay rate of $0.042\% \text{ h}^{-1}$). After 10000 cycles' ADT, Ir_{NC}-Mo₂C/C suffered from a decay of 45 % in j_0 (almost no decay for Ir_{SA}-Mo₂C/C).

Figure R10. (a) Polarization curves of the catalysts in H₂-saturated 0.1 M KOH solution with the rotation rate of 1600 rpm. (b) Tafel plots of the catalysts where the current densities were normalized to their ECSA by CO-stripping voltammetry. (c) The chronoamperometry at 50 mV of Ir_{NC}-Mo₂C/C tested in H₂-saturated 0.1 M KOH with a rotation rate of 900 rpm. (d) Polarization curves for Ir_{NC}-Mo₂C/C before and after ADT test with the rotation rate of 900 rpm.

Revision: Page 6, “And the Ir_{SA}-Mo₂C/C also displayed higher HOR activity than Ir_{NC}-Mo₂C/C, indicating the benefit of the monoatomic dispersed Ir in Mo₂C.”

Page 9, “While for the Ir_{NC}-Mo₂C/C, Ir/C, Pt/C and PtRu/C, the current density remained only 62%, 45%, 35% and 33% after a short test of 20 h, respectively. The decay rate of Ir_{SA}-Mo₂C/C was 0.042% h⁻¹, much smaller than those of Ir_{NC}-Mo₂C/C (1.9 % h⁻¹), Ir/C (2.5 % h⁻¹), Pt/C (2.9 % h⁻¹) and PtRu/C (3.0 % h⁻¹), showing nearly two order of magnitude slowdown of the catalyst decay.”

Page 10, “Ir_{SA}-Mo₂C/C maintained its activity in the whole process of ADT without any decline, while PtRu/C, Pt/C, Ir/C and Ir_{SA}-Mo₂C/C suffered from a decay of 65%, 76%, 14% and 45% in j_0 after scanning for 10,000 cycles, respectively.”

Comment 4. *The Ir-N₄C monodisperse catalysts were not prepared and tested in the experiment. Therefore, it is not necessary to use DFT calculations to compare IrMo₂C and IrN₄C models. It is suggested to compare the activity of atomically dispersed and cluster dispersed Ir-Mo₂C catalysts using theoretical calculation.*

Response: We synthesized Ir_{SA}-N₄-C and its HOR performance was shown in Supplementary Fig. 9 in the original manuscript. The Ir_{SA}-N₄-C showed negligible HOR activity, which was consistent with the reports (*Nat. Catal.* 2023, 6, 916).

According to the suggestion by the reviewer, we calculated the energies of the intermediates on Ir_{NC}-Mo₂C/C catalyst. And the results are shown in Figure R11.

Ir_{NC}-Mo₂C shows a stronger H adsorption with $\Delta G_H = -0.40$ eV than Ir_{SA}-Mo₂C, indicating the monoatomic dispersed Ir has an optimized H adsorption. Ir_{NC}-Mo₂C also exhibits an energy barrier of 0.80 eV with the water generation step as potential-determining step, which is higher than that of Ir_{SA}-Mo₂C.

Figure R11. (a) The models of the Ir_{NC}-Mo₂C and with adsorbed intermediates. (b) The Gibbs free energy diagram of HOR on the Ir_{NC}-Mo₂C.

Revision: Page 11, “When an Ir cluster was placed on the Mo₂C (101) substrate (Ir_{NC}-Mo₂C), a stronger H adsorption with $\Delta G_H = -0.40$ eV was found, indicating the monoatomic dispersed Ir optimized the H adsorption.”

Page 12, “The Gibbs free energy profiles for Ir_{NC}-Mo₂C, Mo₂C (101) and IrO₂ (110) were calculated as well (Supplementary Fig. 24-26), and they showed larger energy barriers of 0.80, 1.57 and 1.03 eV, respectively.”

Comment 5. *Fig. 5 shows that the Gibbs free energy of H on the Ir site of Ir-Mo₂C is much larger than that of platinum (Pt) (about -0.15 eV). However, Ir-Mo₂C has higher HOR activity than Pt/C. H binding energy cannot explain the enhanced activity. How to understand the improved activity? In addition, it is unclear whether the active site for HOR in Ir-Mo₂C is Ir or Mo. In order to confirm the reaction mechanism and enhanced catalytic activity and stability, the optimal active sites for H and OH adsorption, the co-adsorption of H and OH, and the whole HOR mechanism need to be calculated and determined.*

Response: We thank the reviewer for the valuable question and suggestion. The Gibbs free energy of H on Ir_{SA}-Mo₂C was -0.25 eV, which is located in the optimized region, but slightly stronger than that of H on Pt. The possible reason for the better HOR activity of Ir_{SA}-Mo₂C is the enhanced OH adsorption compared with Pt. In alkaline condition, the OH adsorption was found important for HOR kinetics. Markovic et al. considered that the *OH combined with *H to generate water through the bifunctional mechanism, and thus accelerating the H desorption (*Nat. Chem.* 2013, 5, 300). While Chen et al. proposed that *OH helped the formation of the stronger H-bond network among interfacial water molecules, which was important for the H transfer (*Nat. Catal.* 2022, 5, 900). Both possible mechanisms suggested that the enhanced OH adsorption could improve HOR. Ir has enhanced OH adsorption than Pt (*Nat. Chem.* 2013, 5, 300). The calculation results demonstrate that the Ir_{SA}-Mo₂C has even stronger OH adsorption than Ir, which significantly improved the HOR activity. And the discussion about the OH adsorption was included in Page 12 of revised manuscript.

According to the reviewer’s suggestion, we calculated the H and OH adsorption on different sites. We put the H or OH on the Ir site and 5 different Mo sites (illustrated in Figure R12), and then relaxed them to get the released states (summarized in Figure R13-14 and Table R3). It was found that the Ir-Mo hollow site showed the optimized H adsorption Gibbs free energy of -0.25 eV. For the OH adsorption, it was favored on the Mo sites.

Figure R12. The atomic structures of Ir_{SA}-Mo₂C.

Figure R13. The atomic structures of Ir_{SA}-Mo₂C with adsorbed H on different sites.

Figure R14. The atomic structures of Ir_{SA}-Mo₂C with adsorbed OH on different sites.

Table R3. Gibbs free energy of H and OH in different active sites.

Initial site	H adsorption		OH adsorption	
	Final site	ΔG_H (eV)	Final site	ΔG_{OH} (eV)
Mo site-1	hollow site	-0.25	Mo site-1	0.01
Mo site-2	bridge site	-0.41	Mo site-2	-0.60
Mo site-3	bridge site	-0.41	bridge site	-0.61
Mo site-4	Mo site-4	-0.50	bridge site	-0.61
Mo site-5	Mo site-5	-0.61	Mo site-5	-0.70
Ir site	bridge site	-0.41	Mo site-1'	-0.05

We further calculated the whole HOR mechanism through four reaction paths with different active sites. The Gibbs free energy diagrams are shown in Figure R15. The water generation step is the potential determining step of these four paths. And path 1 shows the lowest energy barrier of 0.20 eV. We have added these new findings to the revised manuscript and SI.

Figure R15. The Gibbs free energy diagrams of HOR on the Ir_{SA}-Mo₂C through four different paths and the corresponding atomic structures of the intermediates.

Revision: Page 11, “The H adsorptions on the different sites of Ir_{SA}-Mo₂C were evaluated (Supplementary Fig. 21 and Supplementary Table 6), and it was found that the Ir-Mo hollow sites offer the suitable ΔG_H of -0.25 eV.”

Page 12, “Based on the screening of the different sites of Ir_{SA}-Mo₂C, it was found that the Mo sites were more favored to adsorb OH (Supplementary Fig. 22 and Supplementary Table 6).”

Page 12, “Four reaction paths were considered on Ir_{SA}-Mo₂C through different adsorption sites (Supplementary Figure 23). Because of the enhanced OH adsorption on Mo sites, all the paths exhibited lower energy barrier for OH⁻ adsorption step, giving the potential determining step of water generation step (i.e., $*H + *OH \rightarrow *H_2O$) on Ir_{SA}-Mo₂C. The favourite path showed a lower energy barrier of 0.20 eV, indicating its fastest HOR kinetics.”

Comment 6. Please provide the side view of the corresponding models, and the cell structure and cell parameters of both Mo₂C and Ir-Mo₂C models. To validate the calculation model and reaction mechanism, it is recommended to provide the electronic structure of the corresponding crystal

surface (such as Bader charge, pdos, etc.) for comparison with the corresponding valence data in experiments.

Response: Thanks for the valuable notification and suggestion. The side view of the models and the cell parameters have been added to the revised SI.

According to the reviewer's suggestion, we calculated the Bader charge of the surface atoms in the Mo₂C (101) and Ir SA-Mo₂C. Figure R16 shows the results of Bader charge of these atoms. When Ir was introduced, the Bader charge on Mo adjacent to Ir increased, implied the transferred electrons. And this is consistent with the Mo 3d XPS results, where the proportion of Mo⁶⁺ species increased when Ir introduced. The DOS and PDOS analysis were provided in the original manuscript (Figure 5a) and SI (Supplementary Figure 20). It shows that the Ir-5d orbitals of Ir_{SA}-N₄-C are very localized. By contrast, the Ir-5d orbitals of the Ir_{SA}-Mo₂C exhibit a large degree of delocalization, suggesting its metal-like electronic structure of Ir atoms in Ir_{SA}-Mo₂C, which is similar to the 5d bands of Ir (111).

Figure R16. Bader charge of Ir_{SA}-Mo₂C and Mo₂C (101).

Revision: Page 10, “Bader charge analysis (Supplementary Fig. 19) suggests the electron transfer from surface Mo to the adjacent Ir in Ir_{SA}-Mo₂C, consistent with the XPS results.”

Comment 7. Can the monoatomic dispersed Ir-Mo₂C/C catalyst resist CO poisoning?

Response: We thank the reviewer for the valuable question. We have tested the Ir_{SA}-Mo₂C/C catalyst in 1000 ppm CO/H₂ saturated 0.1 M KOH by using chronoamperometry method at 0.1 V (Figure R17). Ir_{SA}-Mo₂C/C showed enhanced anti-CO poisoning ability compared with Ir/C and PtRu/C. However, a notable decay of 19% was observed after 1500 s of test.

Figure R17. The relative current density at 100 mV of Ir_{SA}-Mo₂C/C, Ir/C and PtRu/C in 1000 ppm CO/H₂-saturated 0.1 M KOH using chronoamperometry technique and RDE method with the rotation rate of 900 rpm. The catalyst loading is about 7 $\mu\text{g}_{\text{PGM}} \text{cm}^{-2}$.

Reviewers' Comments:

Reviewer #1:

Remarks to the Author:

The authors have addressed the comments appropriately. The manuscript quality was improved after revision, and is now in good form for publication.

Reviewer #2:

Remarks to the Author:

The authors have undertaken considerable efforts to strengthen the claims presented in the manuscript and I can now recommend this manuscript for publication in Nat. Commun.

Reviewer #3:

Remarks to the Author:

The authors have addressed the questions asked in the previous round well, and the revised manuscript is now appropriate for publication.